# Effect of Polymer Composition on the Optical Properties of a New Aggregation-Induced Emission Fluorophore: A Combined Experimental and Computational Approach

**DOI:** 10.3390/polym15173530

**Published:** 2023-08-24

**Authors:** Alberto Picchi, Qinfan Wang, Francesco Ventura, Cosimo Micheletti, Jesse Heijkoop, Francesco Picchioni, Ilaria Ciofini, Carlo Adamo, Andrea Pucci

**Affiliations:** 1Department of Chemistry and Industrial Chemistry, University of Pisa, Via Moruzzi 13, 56124 Pisa, Italy; alberto.picchi@phd.unipi.it (A.P.); cosimo.micheletti@phd.unipi.it (C.M.); ventura.francesco89@gmail.com (F.V.); 2Institute of Chemistry for Life and Health Sciences (i-CLeHS), École Nationale Supérieure de Chimie de Paris, PSL Research University, Centre National de la Recherche Scientifique (CNRS), FRE2027, 11, rue Pierre et Marie Curie, F-75005 Paris, France; qinfan.wang@chimieparistech.psl.eu (Q.W.); ilaria.ciofini@chimieparistech.psl.eu (I.C.); 3Department of Chemical Engineering, Product Technology, University of Groningen, 9747 AG Groningen, The Netherlands; heijkoopnj@gmail.com (J.H.); f.picchioni@rug.nl (F.P.)

**Keywords:** push–pull tetraphenylethylene fluorophore, aggregation-induced emission, density functional theory investigations, methacrylate copolymers, quantum yield (TPE), luminescent solar concentrator, optical and device efficiencies

## Abstract

Nowadays, fluorophores with a tetraphenylethylene (TPE) core are considered interesting due to the aggregation-induced emission (AIE) behavior that enables their effective use in polymer films. We propose a novel TPE fluorophore (***TPE-BPAN***) bearing two dimethylamino push and a 4-biphenylacetonitrile pull moieties with the typical AIE characteristics in solution and in the solid state, as rationalized by DFT calculations. Five different host polymer matrices with different polarity have been selected: two homopolymers of poly(methylmethacrylate) (PMMA) and poly(cyclohexyl methacrylate) (PCHMA) and three copolymers at different compositions (P(MMA-co-CHMA) 75:25, 50:50, and 25:75 mol%). The less polar comonomer of CHMA appeared to enhance ***TPE-BPAN*** emission with the highest quantum yield (QY) of about 40% measured in P(MMA-co-CHMA) 75:25. Further reduction in polymer polarity lowered QY and decreased the film stability and adhesion to the glass surface. LSC performances were not significantly affected by the matrix’s polarity and resulted in around one-third of the state-of-the-art due to the reduced QY of ***TPE-BPAN***. The theoretical investigation based on density functional theory (DFT) calculations clarified the origin of the observed AIE and the role played by the environment in modulating the photophysical behavior.

## 1. Introduction

Photovoltaic (PV) energy harvesting is not always cost-effective due to efficiency losses of the panels when not directly illuminated and facing south [1]. Various technologies were developed to enhance photovoltaic energy harvesting or to reduce its cost. Organic fluorophores are generally used for energy harvesting applications, e.g., in luminescent solar concentrators (LSCs), in which they are embedded in a transparent polymeric matrix. When hit by radiation, both indoor and outdoor fluorophores can absorb photons and exploit a fluorescence emission [2]. An ideal fluorophore should have a broad absorption spectrum and a high photoluminescence quantum yield (QY) to harvest and emit the highest number of photons. Moreover, a small overlap between the absorption and emission spectra is preferred to limit self-absorption effects, and the fluorescence quenching phenomenon must be reduced to avoid efficiency losses. For instance, aggregation-caused quenching (ACQ) is one of these phenomena [3], common in the condensed phase for organic fluorophores. A particular class of fluorophores exhibits the opposite behavior: aggregation-induced emission (AIE) or aggregation-caused enhanced emission (AIEE) [4]. When packed, these molecules undergo different mechanisms resulting in a relevant suppression of the non-radiative relaxation paths and leading to the observed bright solid-state fluorescence. A typical AIEgen is tetraphenylethylene (TPE) [5,6], for which a restriction of intramolecular motions responsible for the non-radiative decay going from solution to aggregate phases is often considered as the case of the observed AIE/AIEE. TPE is easily functionalised with donor and acceptor moieties to modulate their photophysical properties, and various TPE derivatives were studied in previous works [7,8,9,10] for LSC applications. Applied in the solid state, the fluorophores in the photovoltaics should boost the QY, thus AIEgens represent the ideal emitters in LSCs. For example, a series of AIE fluorophores (bearing nitril and aromatic moieties) have already been synthesized to meet the requirements of large Stokes shifts and high QY in LSCs [11]. In this paper, we report the synthesis and characterization of a new TPE derivative bearing two dimethylamino push groups and a 4-biphenyl acetonitrile pull group (hereafter ***TPE-BPAN***) as schematically depicted in Figure 1.

The two donors and the acceptor moieties induce a shift of the emission maximum towards longer wavelengths with respect to the native TPE, in agreement with the literature [12]. The enhancement of emission upon aggregation both in solution and solid state has been rationalized on the basis of the results of DFT calculations. In addition to these computational studies, characterization of optical behavior both in solution and in a polymeric matrix was performed. A glassy polymeric surround prevents the rotational motion of fluorophore molecules, as occurs in the solid state [13]. For ***TPE-BPAN*** we chose poly(methyl methacrylate) (PMMA) as the host material, which is widely used in optical applications due to its high transparency and internal transmittance [14,15]. In order to enhance polymer–fluorophore compatibility, PCHMA has already been used by our group in other publications [16,17,18,19]. PCHMA increased the compatibility of the fluorophore, but its lower polarity compromised the time stability of the films. Therefore, we aimed to determine the trade-off between fluorophore performances and polarity by preparing copolymers at different CHMA contents (25, 50, and 75 mol%). QY was measured to determine the best polymer matrix for ***TPE-BPAN*** and its feasibility in LSC applications [20]. Finally, LSC performances were evaluated in terms of photonic (both external η_ext_ and internal η_int_) and device (η_dev_) efficiencies.

## 2. Materials and Methods

All the experimental details are available in the Appendix A section.

## 3. Results and Discussion

### 3.1. TPE-BPAN in Solution

First, the ***TPE-BPAN*** was synthesized through a McMurry reaction between bis-(dimethylamino)benzophenone and bromobenzophenone in the presence of TiCl_4_ and zinc powder, forming the TPE core [21]. The product was then converted by formylation with dimethylformamide (DMF) into a TPE-CHO derivative, to which 4-biphenyl acetonitrile was added, yielding ***TPE-BPAN*** by a Knoevenagel-type condensation. 

The absorption spectra of ***TPE-BPAN*** in tetrahydrofuran (THF) solution were then recorded at increasing concentrations of ***TPE-BPAN*** spanning the range of 10^−6^–10^−5^ M, as shown in Appendix A. All spectra are characterized by a low-absorption peak at 278 nm and two broad and intense absorption bands at 339 nm and 444 nm, respectively, with their maxima unaffected by concentration. Regarding emission properties, ***TPE-BPAN*** is not fluorescent in pure THF solution (Figure 2).

Conversely, a well-defined fluorescence peak appears at 639 nm by adding more than 60% of water, a solvent in which ***TPE-BPAN*** is insoluble, thus suggesting the AIE character of the molecule [22,23]. Interestingly, the resulting Stokes shift is around 200 nm, which is remarkably promising for LSC applications [24,25]. Aggregated ***TPE-BPAN*** particles from a 10/90 vol.% THF/H_2_O mixture were also characterized by dynamic light scattering (DLS) measurements and found to have an average diameter of 206 ± 2 nm (see Appendix A).

***TPE-BPAN*** also displayed good thermal stability since the main weight loss in thermogravimetric analysis (TGA) occurred over 300 °C (Appendix A).

To better understand the nature of the emissive species and the electronic origin of the observed AIE phenomena, DFT calculations were performed to characterize this system in solution and in the aggregated phases. Computational details are extensively reported as Appendix A together with the description of the benchmark calculations that were performed to set up adequate models and level of theory.

Let us first focus on the photophysical properties of isolated ***TPE-BPAN*** in solution. The computed absorption and emission energies associated with the lowest energy intense transition in the gas phase, THF, and water at the TD-DFT level are reported in Table 1. 

As more extensively reported in Appendix A (Appendix A, Appendix A, and corresponding discussion), independently of the level of theory tested, the computed absorption spectra are characterized by two bands in qualitative agreement with the experimental observations. If all range-separated approaches tested (Appendix A) significantly overestimate the lowest energy transition, the description provided by the global hybrid PBE0 method [26,27] appears as the closest to the experimental value, predicting an intense absorption in THF at 2.28 eV (Table 1).

This lowest energy band’s energy is practically unaffected by the change of solvent polarity (i.e., in going from THF to water). We should recall here that these calculations were performed including solvent as a continuum polarizable (through the PCM model) [28] and that the calculation performed in water cannot be directly compared to the experimental results obtained for the THF–water mixtures where ***TPE-BPAN*** is experimentally leading to aggregates. The treatment of these latter mixtures will be addressed in the following paragraph.

The lowest energy band stems from a single electronic transition from the ground (S_0_) and the first excited state (S_1_). This transition has a dominant HOMO to LUMO character, and the analysis of the natural transition orbitals (NTO) [29] associated with this excitation (reported in Figure 3) clearly illustrates its intramolecular charge transfer (CT) nature. The CT is indeed occurring from the donor dimethylamino groups (hereafter labelled as A1/A2, refer to Figure 1 for labelling) to the phenyl of the TPE core linked with the acceptor unit, with a marked directional donor to acceptor character, which breaks the symmetry of the TPE core. Thus, any structural modification modulating the conjugation of these peripheral groups is expected to strongly affect the transition energy.

A quantification of the hole–electron separation at the ES can be provided by the D_CT_ index [30,31] which measures the distance between the barycenters of the computed hole and the electron charge distributions. For the CT transition, a very large charge transfer length is computed with an associated D_CT_ of 7.46 Å. The barycenters of the hole and electron charge distributions are also illustrated in Figure 3.

Due to the relevant donor to acceptor CT character of this transition, a significant relaxation is expected to occur in the excited state leading to a sizeable Stokes shift. Indeed, and in very good agreement with the experiment, a difference of roughly 0.7 eV is computed between absorption and emission energies in THF solution. As can be noted from Table 1, this Stokes shift is also not strongly dependent on the polarity of the environment. From a structural point of view, relaxation in the excited state, as expected, mainly involves a planarization of the substituents around the central TPE core, enhancing their electronic coupling. This effect can be qualitatively inferred from Appendix A where the ground and first excited state-optimized structures of ***TPE-BPAN*** have been superposed. Overall, the most relevant changes are related to the planarization of the peripheral phenyl carrying the A1 and A2 donor groups and of the biphenyl group present in the acceptor substituent. A more quantitative analysis of the relaxation process is provided in SI (Appendix A) for the interested reader.

Focusing now on the intensity predicted for the emission, we can note that a sizable fluorescence would be expected in THF from the data reported in Table 1, in clear disagreement with the faint emission experimentally recorded in pure THF solution. As previously reported for TPE derivatives [18], this discrepancy is related to the neglection of non-radiative decay channels mediated by the vibrational degrees of freedom that have been proven to be particularly effective for TPE and TPE derivatives. To indirectly assess their relevance, an analysis of the computed Huang–Rhys (HR) factors [32] was thus performed. HR factors are obtained by comparing relaxed molecular geometries and normal modes at the ground and excited states. Larger HR factors imply high efficiency of the non-radiative channels associated with molecular vibrations.

In a previous work [18], we have already used this analysis to assess the relative importance of non-radiative decay channels in native TPE and in a TPE derivative (***TPE-MRh***) whose only difference with respect to ***TPE-BPAN*** is the presence of a different acceptor (i.e., methyl-rhodanine). We can thus qualitatively compare the HR factors computed here for ***TPE-BPAN*** to those reported in our previous work computed at a similar level of theory to estimate the efficiency of non-radiative vibrational channels.

The three largest HR factors computed for ***TPE-BPAN*** (Appendix A) are all associated with low frequencies, namely to the 3rd, 4th, and 5th normal modes, with frequencies computed at 14, 20, and 24 cm^−1^, respectively. These HR factors (12.8, 6.9, and 3.2, respectively) are smaller than those computed for TPE (whose emission is totally quenched by non-radiative decay) but comparable to those computed for ***TPE-MRh***.

The analysis of these three normal modes shows that they all involve a structural reorganization corresponding to the rotation of donor and acceptor peripheral groups as schematically depicted in Figure 4. Nonetheless, and as already pointed out for ***TPE-MRh***, these normal modes are less coupled than for native TPE and, thus, a less pronounced structural reorganization and a less efficient activation of the non-radiative channels is expected with respect to unsubstituted TPE. We can thus conclude that, analogously to ***TPE-MRh***, in the case of ***TPE-BPAN***, non-radiative decay is also responsible for the observed faint fluorescence in THF solution and mixtures where this molecule is soluble, although this non-radiative pathway is unable to completely suppress emission in solution as in the case of unsubstituted TPE.

### 3.2. TPE-BPAN Crystalline Aggregates

The XRD pattern experimentally recorded (Appendix A) is consistent with the formation of polycrystalline particles of ***TPE-BPAN*** in water/THF mixtures containing 70% (or more) of water. The three most thermodynamically stable polymorphs (hereafter labelled as Pol_1, Pol_2, and Pol_3), predicted following the computational procedure detailed in Appendix A, are showing some of the characteristic peaks experimentally observed in the XRD pattern, in particular those reported at 4.002°, 6.599°, 8.099°, 13.806°, and 16.220° (Appendix A). Pol_1, Pol_2, and Pol_3 can thus be considered as realistic estimates of the most largely occurring polymorphs present in the particles aggregating from water/THF mixtures. More information on their structural properties can be found in Appendix A.

In order to compute the solid-state absorption and emission energies of each of these polymorphs, the Ewald embedding procedure, previously developed and applied by some of us for the calculation of organic crystalline phases, was applied (refer to Appendix A for details) [33,34,35,36,37]. The computed absorption and emission properties are summarized in Table 2, together with relevant experimental data.

In agreement with the experimental observations, absorption and emission energies are only marginally shifted going from THF solution to crystalline phases. The nature of all these transitions is not changing from solution, still corresponding essentially to a HOMO–LUMO transition with relevant implication of the donor and acceptor groups, although the analysis of the D_CT_ index computed for the polymorphs shows a sizable reduction of the CT extent (of roughly 1.5 Å on average). Experimentally, a redshift of roughly 0.1 eV is observed both for absorption and emission, while computationally a 0.1 eV blueshift is computed for absorption and a negligible shift in emission is computed for the most stable polymorph, which is acceptable agreement considering the accuracy expected from TD-DFT calculations. More interestingly, considering the computed oscillator strengths, one can note that a sizable fluorescence is expected for all polymorphs, with computed oscillator strengths ranging from 0.154 to 0.255 a.u., and that these computed intensities are comparable to those computed in THF solutions.

The observed gain in fluorescence upon aggregation must thus be due to the deactivation of the non-radiative de-excitation channels effective in solution due to the presence of relevant intermolecular interactions induced by the crystalline packing of the different polymorphs. These intermolecular interactions must hinder the vibrational motions mainly involving the torsion of the donor and acceptor moieties with respect to the central TPE core that were previously identified from the analysis of the HR factors computed in solution as the most relevant non-radiative de-excitation channels.

To detect intermolecular interactions in the crystalline environment, the interaction region indicator (IRI) analysis [38], as implemented in the Multiwfn package [39], was applied as reported in Appendix A. Several interactions can be highlighted. For instance, in Pol_2, C-H···π· interactions exist between the A1 and P2 groups, which restrict the rotation of the P2 part. The A2 group is involved in C-H···N and C-H···H van der Waals interactions with nitrile and the phenyl linker, which restrict the vibrational degrees of freedom of both the nitrile and the phenyl involved. Finally, in the Pol_3 system, the intermolecular C-H···π interactions appear between the A1, A2 and P1, P2 groups.

Further analyses indicate that, especially in Pol_3, relevant intermolecular packing between the donor and acceptor groups of adjacent molecules can significantly restrict their freedom, de facto de-activating the non-radiative channels and leading to the observed enhancement in emission upon aggregation.

The structural constraint induced by the crystal packing is also indirectly probed by comparing the structural relaxation upon excitation occurring in solution and in the different polymorphs. Appendix A monitors this variation by analyzing the most relevant dihedrals. Of note, the relaxation involving the coupling of the acceptor and donor unit with the central core of the molecule is always smaller in the crystalline phases, thus confirming the constraints imposed by the crystal packings. On the other hand, the relaxation of the donor unit, and particularly that of the P1-P2 biphenyl moiety, is predicted to be larger in the crystalline phases but is not expected to provide significant non-radiative decay channels since these groups are not massively involved in the vibrational frequencies emerging as relevant for de-excitation from the HR factors analysis.

### 3.3. TPE-BPAN in Polymeric Thin Films

***TPE-BPAN*** was next investigated in polymeric thin films. These were prepared by means of the solvent casting technique on a 50 × 50 × 3 mm^3^ glass with high optical purity acting as a waveguide. The polymer selected was initially poly(methyl methacrylate) (PMMA), which has been widely used in LSC systems. Subsequently, ***TPE-BPAN*** was investigated in other polymer matrices, such as poly(cyclohexyl methacrylate) (PCHMA), to assess its performance in matrices of different polarity. It had already been observed that a less polar polymer such as PCHMA could lead to higher QY [16]; however, the high fragility of PCHMA in contact with glass compromises its long-term use. 

We then synthesized five different polymeric matrices by free radical polymerization (FRP), i.e., the two homopolymers poly(methyl methacrylate) (PMMA) and poly(cyclohexyl methacrylate (PCHMA), and three copolymers obtained from the polymerization of methyl methacrylate and cyclohexyl methacrylate at different compositions (P(MMA-co-CHMA), 75:25, 50:50, and 25:75 mol.%). All polymerizations were conducted in toluene, using 1 wt.% AIBN as an initiator and adding monomers in the desired composition. The polymerizations required from 24 to 72 h to achieve complete conversion, verified with ^1^H NMR spectroscopy (Appendix A). FTIR showed the typical absorption of methacrylate polymers and with only small changes due to the different alkyl contribution near 2900 cm^−1^ (Appendix A). The obtained polymers were characterized in terms of molecular weight and polydispersity index (PDI) by gel permeation chromatography (GPC). Despite variations in the polarity of the comonomers and in the composition of the reaction mixtures, no substantial differences were observed between the five polymers synthesized, with the average molecular weight (M_w_) ranging from 35′000 to 95′000 and the PDI ranging from 1.5 to 2.8 (Appendix A). Differential scanning calorimetry (DSC) measurements showed a drop in the glass transition temperature (T_g_) with increasing CHMA content, with no noticeable relationship with the average molecular weight (Appendix A). Notably, the copolymers’ T_g_ agrees with the Fox equation (R^2^ = 0.983). All polymer matrices showed a suitable thermal stability for outdoor applications, as the depolymerization reaction, which causes the main weight loss in a thermogravimetric analysis (TGA), occurred over 200 °C.

The polarity of the matrices was verified by determination of the static contact angle, resulting in 70° ± 2 for PMMA, 91° ± 2 for P(MMA-co-CHMA) 50:50, and 99° ± 2 for PCHMA (Appendix A). 

As shown in Figure 5, the first peak at 338–340 nm showed no change in position for all thin films compared to the analysis in THF solution. Notably, the second peak, which was centered at 444 nm in THF, is slightly shifted to 434–439 nm in polymeric thin film, apart from an evident variation to 427 nm in the PCHMA homopolymer matrix. As discussed above, the electronic transition responsible for the lowest energy band is actually a CT from the donor to the acceptor. By inspection of the computed data reported in Table 1, one can notice that while no difference can be remarked when moving from water to THF, when moving further to a fully non-polar environment (gas phase), a shift to higher energy is predicted. This agrees with a solvatochromic effect induced by the presence of the remarkably less polar matrix of PCHMA [40].

***TPE-BPAN*** emissions peaks are all included in the 570–600 nm range (Figure 5), i.e., in a range where the external quantum efficiency of the Si-PV cell is at a maximum. As can be seen from Figure 5, referring to the 0.4 wt.% concentration, the wavelength at the emission maximum does not show any relevant shift as the CHMA content in the polymeric matrix increases, except for the 100% PCHMA sample, where a blue shift is observed from 575 to 571 nm. More evident differences are recognized at the 2.0% concentration. In fact, although a red shift caused by self-absorption was expected as the fluorophore concentration increased, a red shift of 15 nm was observed in the matrices with a high CHMA content. The lower polar character of the PCHMA matrix is probably responsible for these different shift effects. It is to be noted that a blue shift of the emission peak with increasing CHMA content leads to a reduction of the Stokes shift, which may result in a more pronounced self-absorption effect.

The trends in terms of QY of the thin films at different monomer compositions and fluorophore concentrations are shown in Figure 6. As can be seen, the presence of CHMA in the copolymer appeared to have a positive effect on enhancing the fluorophore’s emission. In particular, ***TPE-BPAN*** displayed a higher PLQY at all concentrations studied when dispersed in P(MMA-co-CHMA) 75:25. Above 25% CHMA content, a decrease in QY was observed until values almost identical to those of ***TPE-BPAN***/PMMA films, which was also possibly caused by the increased self-absorption due to the Stokes shift reduction. Also confirmed in this work is the result reported earlier [18], where a similar ACQ phenomenon was observed in polymer films, and the behavior was attributed to the formation of less emissive amorphous aggregates. The presence of such aggregates, increasing in size with concentration from 0.4 to 2.0 wt.%, was observed using epifluorescent microscopy (Appendix A).

Since films of P(MMA-co-CHMA) 75:25 and PMMA showed good integrity and adhesion to the glass substrate, they were investigated as matrices in LSC applications, and the main results were determined according to recently published laboratory protocols [41,42]. Figure 6 compares the internal (η_int_) and external (η_ext_) photonic efficiencies of LSC made from these two matrices. As was the case for the quantum yield, the copolymer-based film results in a higher η_int_ even though this gap tends to zero with increasing ***TPE-BPAN*** concentration. This could be due to a lack of homogeneity in highly concentrated films where fluorophore aggregates could act as scattering centers [43]. Conversely, η_ext_ appears primarily unaffected by the change in matrix polarity since it goes from 2% to about 3% in both cases. These values are considerably lower than those obtained with Lumogen F Red 305 (a state-of-the-art fluorophore for LSC applications) in PMMA thin films, whose efficiencies never fell below 5% in the range of concentrations analyzed. However, the result is not surprising as the quantum yields of Lumogen F Red 305 in PMMA always exceed 70% from 0.4 to 2 wt.% of fluorophore content.

As a final step, LSCs were coupled to monocrystalline silicon photovoltaic cells to measure the electrical device efficiency η_dev_, which is a significant parameter for the characterization of LSC [44]. Again, the results do not differ greatly with 25% of cyclohexyl methacrylate content in the polymer. The percentage of incident solar power converted into electricity always remained between 0.2 and 0.3%, while it ranged from 0.5% (0.4 wt.%) to 1.0% (2.0 wt.%) in the case of Lumogen F Red 305. Thus, matrix polarity and fluorophore concentration variations do not seem to substantially influence the performance of LSC devices.

## 4. Conclusions

A novel AIE fluorophore, ***TPE-BPAN***, was synthesized by functionalizing TPE with two dimethylamino donor groups and a 4-biphenyl acetonitrile acceptor moiety. Its thermal stability was found to be up to 300 °C. In THF solution, two absorption peaks are observed (339, 444 nm), but no emission is detected. A bright fluorescence centered at 639 nm is visible with the addition of water from 70% onwards, thus demonstrating its AIE behaviour. The theoretical calculations demonstrate that the observed enhancement of the emission in aggregate phases can mainly be ascribed to the suppression of non-radiative decay channels that are active in the solution but are ineffective in the crystalline phases due to the constraints imposed by the crystal packings on the movement of donor and acceptor units involved in the absorption and emission processes. This agrees with the behavior observed for TPE and other TPE derivatives, and it is due to several different intermolecular interactions at work in the condensed phases that significantly constrain the torsion of phenyl substituents in the solid state, suppressing non-radiative decay and enhancing the radiative emission process.

The same type of mechanism is active when ***TPE-BPAN*** is embedded in a polymer matrix, where AIE-like emission also occurs. Five polymeric matrices obtained by combining methyl methacrylate as a more polar monomer and cyclohexyl methacrylate as a less polar monomer were proposed. Poly(methyl methacrylate-co-cyclohexyl methacrylate) 75:25 mol.% turns out to be the best polymeric environment for ***TPE-BPAN*** since QY was the highest and thin film integrity and glass adhesion were preserved. LSC characterizations proved that the gain in QY did not significantly enhance the photon concentration effect or the electrical production.

## Figures and Tables

**Figure 1 polymers-15-03530-f001:**
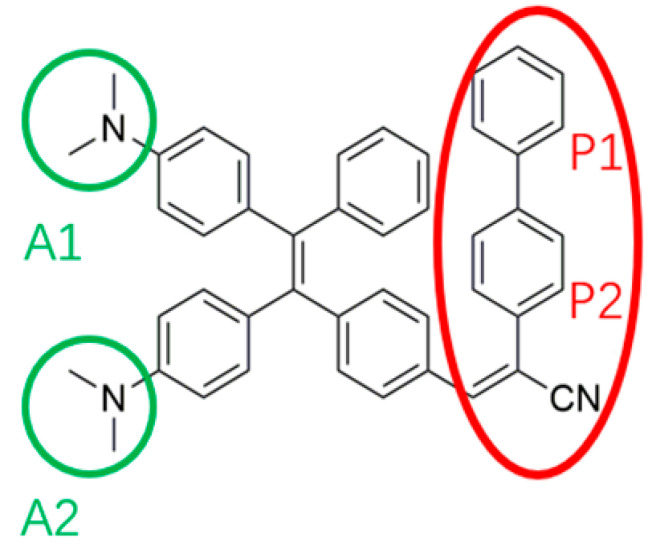
Schematic structure of ***TPE-BPAN*** highlighting the donor (green) and acceptor (red) substituent groups introduced in the TPE core. A1 and A2 represent the amino groups; P1 and P2 are the aromatic rings in the biphenyl group.

**Figure 2 polymers-15-03530-f002:**
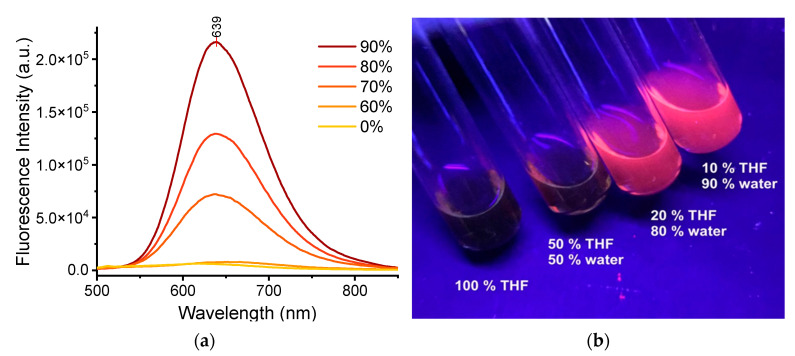
(**a**) Emission spectra of ***TPE-BPAN*** dissolved in THF/H_2_O mixtures at different compositions (% of H_2_O labeled, concentration: 5 × 10^−5^ M); (**b**) Picture showing emission features of ***TPE-BPAN*** in various THF/H_2_O mixtures under UV irradiation (366 nm).

**Figure 3 polymers-15-03530-f003:**
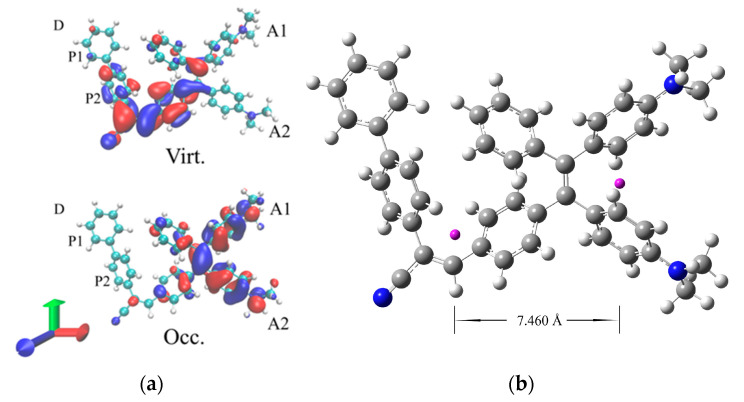
(**a**) NTO orbital associated with the S_0_–S_1_ transition of TPE-BPAN. (isodensity = 0.025 a.u.). (**b**) Graphical representation of the D_CT_: barycenters of hole and electron charge distribution are in violet.

**Figure 4 polymers-15-03530-f004:**
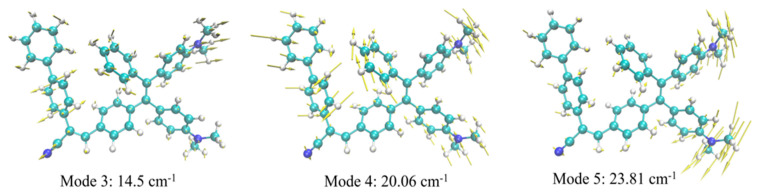
Schematic representation of the normal modes related to the largest HR factors of TPE-BPAN.

**Figure 5 polymers-15-03530-f005:**
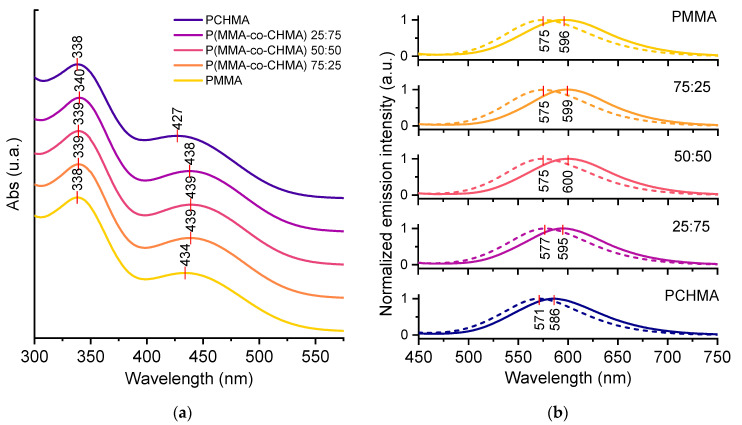
(**a**) Absorption spectra of ***TPE-BPAN*** in the five polymeric matrices under study at 2.0 wt.% concentration. Peak positions are marked with red vertical dashes. (**b**) Emission spectra of ***TPE-BPAN*** in the five polymeric matrices under study in the concentration of 0.4 wt.% (dashed) and 2.0 wt.% (solid). Peak positions are shown with red vertical dashes.

**Figure 6 polymers-15-03530-f006:**
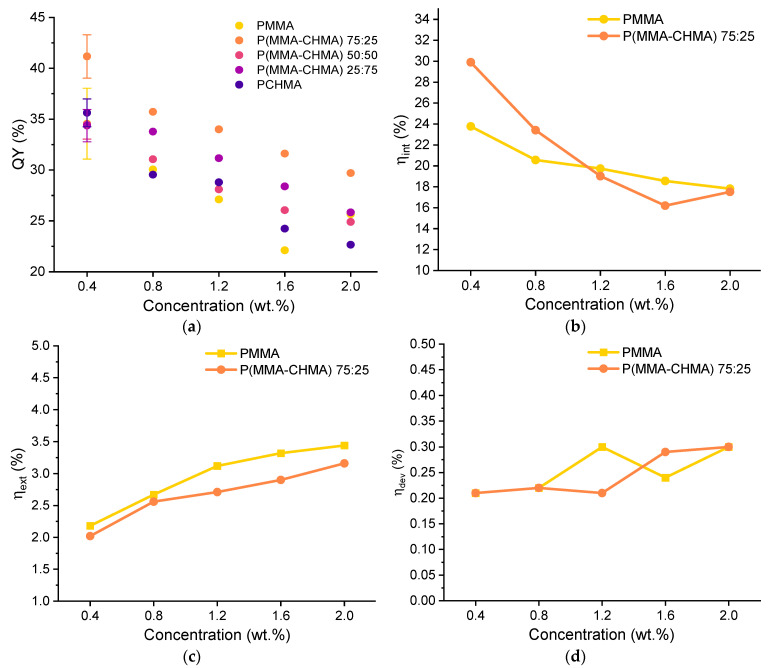
(**a**) Quantum yield (QY %) vs. fluorophore concentration for the five TPE-BPAN-containing thin films. Error bars are shown only with reference to the 0.4 wt.% concentration for clarity (see all data from Figures S28 to S32); (**b**) internal photonic efficiency and (**c**) external photonic efficiency vs. TPE-BPAN concentration in PMMA and P(MMA-co-CHMA) 75:25 films. (**d**) Electrical device efficiency vs. TPE-BPAN concentration in PMMA and P(MMA-co-CHMA) 75:25 films.

**Table 1 polymers-15-03530-t001:** Computed and experimental absorption (E_abs_) and emission (E_emi_) energies of ***TPE-BPAN*** in different environments. Refer to SI for computational details.

	*E*_abs_ (eV)	*E*_em_ (eV)	Emi. Intensity (au)	Stokes Shift (eV)
THF_calc_	2.281	1.578	13,677	0.71
H_2_O_calc_	2.281	1.547	14,413	0.734
gas phase_calc_	2.460	1.789	8675	0.671
THF_exp_.	2.793	2.006	6650	0.787

**Table 2 polymers-15-03530-t002:** Computed absorption and emission energies and oscillator strengths of TPE-BPAN polymorphs in comparison with available experimental data and computed data for TPE-BPAN in THF solution.

		E_abs_ (eV)	f_abs_ (a.u.)	E_emi_ (eV)	f_emi_ (a.u.)
PBE0-D3	Pol_1	2.526	0.255	2.524	0.255
	Pol_2	2.380	0.195	1.767	0.192
	Pol_3	2.459	0.199	1.605	0.154
Exp. 10% THF	-	2.695		1.947	
PBE0	THF sol	2.281	0.290	1.578	0.338
Exp THF solution	--	2.793		2.006	

## Data Availability

Not applicable. No data have been created.

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
