# Peer review of "Effect of Polymer Composition on the Optical Properties of a New Aggregation-Induced Emission Fluorophore: A Combined Experimental and Computational Approach"

_polymers, 2023, doi:10.3390/polym15173530_

Round 1

Reviewer 1 Report

1.     Please improved the keywords, it is not better.

2.     The abstract section is tedious. It is suggested to reorganize the language.

3.     Please maintain a consistent style for the references throughout in the reference List according to the requirements of the journal.

4.     The caption of Fig. 2 should be expressed in detail.

5.     Could you provide the SEM of TPE-BPAN in polymeric thin films?

6.     The characterization of the samples should be boosted. Such as IR, and TGA.

Author Response

We wish to thank the reviewer for the constructive comments that provided valuable insights to refine the manuscript contents and analysis. According to the suggestions, we have thoroughly revised our manuscript; its final version is enclosed. Point-by-point responses to the reviewers’ comments are listed below.

  1. Please improved the keywords, it is not better.

The keyword list was improved.

  1. The abstract section is tedious. It is suggested to reorganize the language.

The reviewer was right. We restyled the abstract to make it also clearer.

  1. Please maintain a consistent style for the references throughout in the reference List according to the requirements of the journal. (Picchi)

We thank the reviewer for pointing this out. We updated the reference list according to the style of the journal.

  1. The caption of Fig. 2 should be expressed in detail. 

Amended.

  1. Could you provide the SEM of TPE-BPAN in polymeric thin films?

Since the TPE-BPAN molecules and aggregates are luminescent when embedded in polymers, we preferred to complete the epifluorescence micrographs for all the prepared samples, as reported in the SI file. In our opinion, this way better evidenced of possible phase separation between the fluorophore and the polymer matrix and the size of the aggregates.

  1. The characterization of the samples should be boosted. Such as IR, and TGA. 

We added FTIR to the SI file and reported a sentence concerning the thermal stability of the polymers and copolymers. All the prepared polymer matrices were stable up to 200 °C and therefore complied with the requirements as polymers for LSC applications.

  1. Some work may be considered, such as Molecules 2023, 28, 3282; New J. Chem., 2020, 44, 16265-16268 and Theor. Chem. Acc. 2022, 141, 68 (Picchi)

We added the first one since the most illustrative for the manuscript's topic.

Reviewer 2 Report

The submitted manuscript is of a very high quality and surely deserves to be published in Polymers. The amount of new results is impressive and the scientific level is very high. However, I have also some suggestions and comments for the Authors.

The abstract is too long and too detailed, it should be shortened.

Figure 2, it isn’t entirely clear whether the P2 group is the cyano group or phenyl ring.

Table S2, the quite large difference between the experimental and theoretical values may be caused by the application of quite small basis set. I would strongly recommend application of 6-311++G(d,p), especially since the modeled molecule is not so large so it should be computationally affordable.

Lines 101-102, actually I can see a third peak at around 275 nm, especially at higher concentrations.

PXRD: I’m sure that the sample is crystalline. Have you tried to determine the unit cell dimensions and space group, based on the indexing?

Honestly, I think that at least some of the results should be moved from SI to main part. Especially the figures.

TPE-BPAN crystals section in SI, I guess by stating “polymorph” the Authors mean “polymorph predictor” tool? Also, since the authors had an access to the Materials Studio, why haven’t you used CASTEP instead of CRYSTAL?

To be honest, the results presented in Figure S33 are not very credible. I believe the Authors have properly set up the calculations, but the accuracy of “polymorph” predictor is questionable-after all it evaluates only specific space groups (only 6 of them). Also, during the CRYSTAL optimization of structures generated by polymorph predictor, have you optimized unit cell dimensions as well? This is a very good indicator of the correctness of such prediction (comparison of the unit cell dimensions from polymorph predictor and optimized ones). Also, the predicted unit cells should be included in the SI as cif files.

Figure 8a, why the error bars are only for 04. Wt %?

Author Response

We wish to thank the reviewer for the constructive comments that provided valuable insights to refine the manuscript contents and analysis. According to the suggestions, we have thoroughly revised our manuscript; its final version is enclosed. Point-by-point responses to the reviewers’ comments are listed below.

The submitted manuscript is of a very high quality and surely deserves to be published in Polymers. The amount of new results is impressive and the scientific level is very high. However, I have also some suggestions and comments for the Authors.

The abstract is too long and too detailed, it should be shortened.

We thank the reviewer for pointing this out. We restyled the abstract to make it also clearer.

Figure 2, it isn’t entirely clear whether the P2 group is the cyano group or phenyl ring.

Amended.

Table S2, the quite large difference between the experimental and theoretical values may be caused by the application of quite small basis set. I would strongly recommend application of 6-311++G(d,p), especially since the modeled molecule is not so large so it should be computationally affordable.

We agree that the use of a larger basis is expected to improve the agreement with the experimental data. Nonetheless, we prefer to keep the calculation at the level of theory chosen since:

  • We want to keep a consistent level of theory for the calculations performed on the isolated molecule and the molecular crystal (to allow their comparison). The larger basis set propose is not suitable nor easily computationally affordable for the subsequent calculations performed in condensed (ie solid) state nor for the calculations of excited state relaxation and frequencies (necessary for instance to evaluate HR factors)
  • Stokes shifts (which are a very relevant observable) are actually correctly reproduced at the level of theory used
  • A similar basis set as been previously applied to study on a similar TPE derivative (namely TPE-MRh) and the use of the same basis allows us to compare the results obtained for the two compounds.

Lines 101-102, actually I can see a third peak at around 275 nm, especially at higher concentrations.

We added this peak in the description of the absorption spectra as a function of molecule concentration. Nevertheless, considering its very low intensity, it was not considered in the general discussion of the TPE-MRh optical features.

PXRD: I’m sure that the sample is crystalline. Have you tried to determine the unit cell dimensions and space group, based on the indexing? 

We thank the reviewer for pointing this out. We were actually unable to determine the space group of the polymorphs present in the samples from the experimental XRD and this was actually the reason of the polymorph search performed computationally.

Honestly, I think that at least some of the results should be moved from SI to main part. Especially the figures. 

We moved Figures 4 and 6 to SI to render the manuscript more interesting and readable.

TPE-BPAN crystals section in SI, I guess by stating “polymorph” the Authors mean “polymorph predictor” tool? Also, since the authors had an access to the Materials Studio, why haven’t you used CASTEP instead of CRYSTAL? 

Yes, indeed it is the ‘Polymorph predictor tool’ which is now correctly pointed to in the SI. Thanks for spotting it. 

The CRYSTAL program was used for two main reasons: 1) it allows the use of localized atomic basis functions (that is GTOs) that are of the same quality of those used for the molecules in solution simulated with Gaussian (contrary to CASTEP which uses planewaves) and the comparison of the results and 2) it allows to perform calculations using hybrid functionals thanks to their computationally efficient implementation.

To be honest, the results presented in Figure S33 are not very credible. I believe the Authors have properly set up the calculations, but the accuracy of “polymorph” predictor is questionable-after all it evaluates only specific space groups (only 6 of them). Also, during the CRYSTAL optimization of structures generated by polymorph predictor, have you optimized unit cell dimensions as well? This is a very good indicator of the correctness of such prediction (comparison of the unit cell dimensions from polymorph predictor and optimized ones). Also, the predicted unit cells should be included in the SI as cif files.

We agree with the reviewer that the search was restricted to a limited number of space groups. Nonetheless, the six space groups chosen were selected on the basis of their (high) occurrence in organic material (as previously done for similar TPE derivatives).  The selection of the space groups is thus motivated. Concerning the crystal structures these were fully reoptimized at DFT level (including cell parameters) using the CRYSTAL code at level of theory expected to correctly simulate intermolecular interactions (that is including for instance dispersion corrections).

Furthermore, the data reported in Table S4 refer to structures re-optimized at DFT level using two different functionals. These data are consistently cross validating the structures and also the relative energetic ranking of the polymorphs. In our opinion these data are thus relevant to validate the correctness of the overall procedure used (that is polymorph search using a FF + DFT re-optimization).

As requested, the predicted unit cells obtained from CRYSTAL calculations are now reported in the SI.

We also agree with the reviewer that Figure S33 may be misleadingly interpreted as a simulation of the experimental XRD while it is only presented to show some of the characteristics peaks experimentally observed. Nonetheless, since most probably the experimental same is containing more than one polymorph a direct one to one comparison with the experimental XRD cannot be performed. To avoid erroneous interpretations, we thus prefer to remove this figure from the revised version of the SI.

Figure 8a, why the error bars are only for 04. Wt %?

The caption explains the reason: “Figure 6. (a) Quantum yield (QY %) vs. fluorophore concentration for the five TPE-BPAN-containing thin films. Error bars are shown only with reference to the 0.4 wt.% concentration for clarity (see all data from Figures S28 to S32);”.

Round 2

Reviewer 2 Report

The Authors have answered my questions, revised their work and improved the manuscript. This version can be accepted for publication.